# Data collection and analysis of temperature field simulation of steam boiler for soil steam disinfection

Sipu Pan ⓘ *

College of Transportation Engineering, Nanjing Vocational University of Industry Technology, Nanjing, Jiangsu, China

* 1127196122@qq.com

## Abstract

Soil steam disinfection (SSD) has emerged as a highly promising substitute for methyl bromide fumigation in the management of soil-borne pathogens, nematodes, and weed seeds. In the present study, an innovative steam boiler driven by Helmholtz-type pulse combustors was meticulously engineered to meet the requirements of SSD in horticultural greenhouses. The water within the boiler was partitioned into discrete zones, and a total of 80 temperature sensors were strategically positioned to precisely monitor the temperature fluctuations at specific locations. Leveraging the Natural Neighbor Interpolation method, a comprehensive model of the temperature field within the boiler was developed. The experimental findings demonstrated that at the initial stage, the temperature in the vicinity of the pulse combustors escalated rapidly, while the regions located farther away exhibited a relatively sluggish heating rate. As the heating process progressed, the area of high temperature expanded progressively. After 35 minutes of operation, the majority of the water within the boiler reached temperatures exceeding 89 °C, signifying the generation of saturated steam. Notably, distinct temperature gradients were discerned along different axes and planes, offering valuable insights for the structural optimization of the boiler. In comparison with other steam boiler models, the designed boiler boasted a relatively compact volume, a lightweight empty weight, and an impressively favorable fuel consumption rate per unit steam production, registering at 0.06867 L·kg⁻¹. These results unequivocally highlight the potential of this boiler for efficient SSD applications, thus laying a solid foundation for further research and development in this field.

## 1. Introduction

Soil steam disinfection (SSD) increases the soil temperature and kills harmful organisms through heat exchange between the steam (at a high temperature) and soil particles. SSD is the main method being developed to replace fumigation with methyl bromide for controlling or even eradicating soil-borne pathogens, nematodes, and weed seeds [1–4]. SSD is fast and does not lead to residual effects [5], particularly in deep soil, so the soil can be used as soon as it has cooled to the ambient temperature [6]. SSD was first developed by Frank in 1888 and was first commercialized by Rudd in 1893 [7]. SSD was used extensively in the 1960s. Since then,

**Data availability statement:** All relevant data are within the manuscript and its Supporting information files.

**Funding:** This paper is funded by the Natural Science Foundation of the Jiangsu Higher Education Institutions of China (No. 22KJB210014), the Start-up Fund for New Talented Researchers of Nanjing Vocational University of Industry Technology (No. YK-19-04-03). The funders had no role in study design, data collection and analysis, decision to publish, or preparation of the manuscript.

**Competing interests:** The authors have declared that no competing interests exist.

various SSD methods (e.g., sheet steaming and steaming using fixed tube pipes) have been developed and used in open fields and horticultural greenhouses [8–12].

Bitarafan et al. [13] investigated thermal control of barnyardgrass seeds using a prototype stationary soil steaming device. Seed germination was decreased by 50% when a maximum soil temperature of 62–68 °C was reached and by 90% when a maximum soil temperature of 76–86 °C was reached. Miller et al. [14] found that soil or sand from the surface to deeper layers was heated more thoroughly and rapidly when steam was applied while the soil or sand was being mixed than when steam was applied to the surface of unmixed (i.e., stationary) soil or sand. Raffaelli et al. [15] designed and used a prototype band-steaming machine. A maximum soil temperature of 63 °C at 25 mm deep was achieved with CaO prior to sowing the crop. Fennimore et al. [16] found that steam treatment then adding mustard seed meal (a fertilizer and source of additional organic matter) markedly improved strawberry yields and controlled weeds and pathogens. Kim et al. [17] also used mustard seed meal to decrease the microsclerotia density in *Verticillium dahliae*, the propagule density in *Pythium ultimum*, and the cumulative weed density.

Identifying a suitable source of steam is the main problem needing to be solved when developing a SSD method. Research is currently being performed around the world to improve the effects of SSD and decrease energy consumption, but few of them proceed from the steam boiler. Commercial steam boilers were used directly in some studies [18–20], but there are some problems with such boilers, particularly related to the boilers being heavy and very large. These problems are particularly important when SSD equipment is used in a horticultural greenhouse because of the lack of space. In the study presented here, we designed a steam boiler to supply steam for SSD in horticultural greenhouses. The water in the steam boiler was heated by Helmholtz-type pulse combustors rather than coal, electricity, or diesel. Pulse combustion is a burning process in which some parameters can be changed over time, such as the temperature, pressure [21]. More efficient combustion and lower emissions (by mass fraction) can be achieved in pulse combustion through combustion with acoustic fluctuations than through normal combustion. Acoustic fluctuations increase turbulence and can therefore improve heat and mass transfer by up to a factor of five compared with stationary gas flow [22,23]. Acoustic excitation increases temperature fluctuations and therefore increases the heat transfer rate without increasing the mean temperature, meaning NO emissions by mass fraction are not increased [24].

Pulse combustion was first described by Lord Rayleigh and was used in many applications (e.g., German V-1 "buzz bombs" used in World War II) in the 20th century [24,25]. Pulse combustion applications can be divided into two general types, systems using thermal pulse combustors as efficient heaters and systems using the pulsating high flow-velocity inside or outside the pulse combustor to improve processes such as drying [26–29] and calcining [30–32]. In this study, a pulse combustion-based steam boiler was meticulously designed to capitalize on the enhanced heat transfer efficiency characteristic of pulse combustion. The water within the steam boiler was segmented into discrete regions, and the temperatures at multiple specific points were accurately measured and determined. Subsequently, the temperature field within the steam boiler was mathematically modeled employing the Natural Neighbor Interpolation method. By doing so, the process of water temperature elevation was precisely identified, thereby furnishing crucial data essential for optimizing the structural configuration of the system and augmenting the steam boiler efficiency.

## 2. Materials and methods

### 2.1. Principles of a steam boiler

A steam boiler containing pulse combustors, a water inlet, a steam outlet, and a tank body, as shown in Fig 1a [33], was designed and manufactured. Each pulse combustor comprised

an air inlet, a fuel inlet, a spark plug, a combustion chamber, a tail pipe, a decoupling chamber, and a tail gas outlet. An adjustable carburetor was employed to regulate the flow rates of air and fuel. The fuel utilized in this study was gasoline. During the combustion process, the peak temperature of the tail gas within the combustion chamber and tail pipe could ascend to several thousand degrees Celsius. Consequently, the combustion chamber, tail pipe, and decoupling chamber were submerged in water, with only the air inlet, fuel inlet, and tail gas outlet remaining exposed outside the tank body. Water entered through the inlet located at the bottom of the tank, while steam exited via the outlet at the top. The tail pipe was designed as a triple-helical configuration (Fig 1b) to augment the heat transfer area and mitigate deformation induced by elevated temperatures. Four pulse combustors were incorporated into the system to deliver the requisite power. A three-dimensional representation of the steam boiler is depicted in Fig 1c.

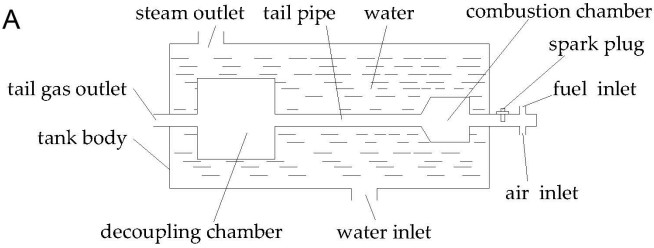

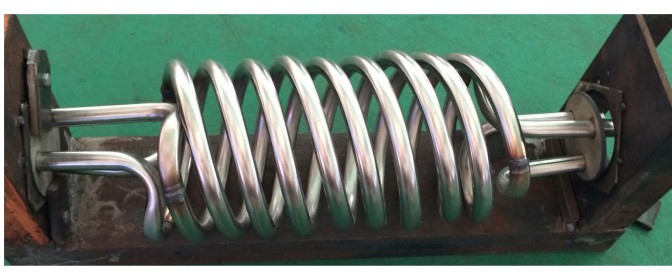

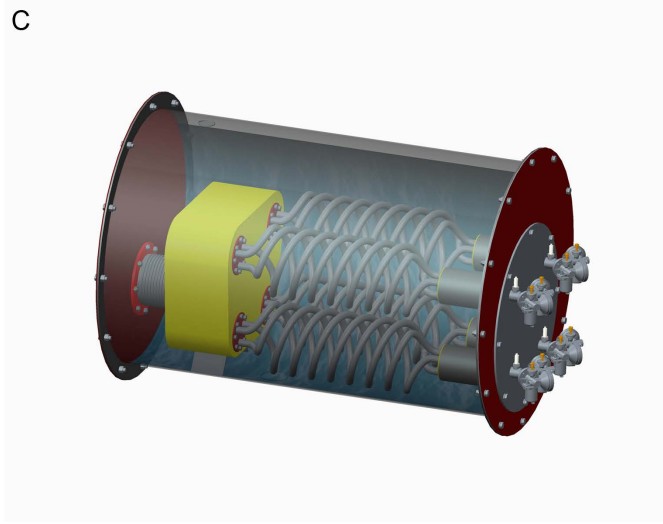

**Fig 1. Steam boiler schematic and the tail pipe.** (a) Schematic diagram of the steam boiler. (b) The spiral pipes. (c) Three-dimensional picture of the steam boiler.

## 2.2. Temperature measuring system

**2.2.1. Temperature measuring devices.** The water temperature was measured using temperature sensors (Shanghai Chi Control Automation Instrument Instruments Co., Shanghai, China). The main parameters for the temperature sensors are shown in Table 1. The temperature data were collected using an intelligent inspection device (Nanjing ChaoYang Instruments Co., Nanjing, China) and recorded using a laptop using XMD-2000A31-2 software (Nanjing ChaoYang Instruments Co.). The main parameters for the intelligent inspection device are shown in Table 2.

**2.2.2. Temperature measuring method.** The temperature of the water contained within the tank body exhibited a continuous upward trend. However, it was infeasible to monitor the water temperature at every single point throughout all time intervals. The optimal approach for gauging the temperature within the system entailed continuously measuring the temperature at the maximum number of accessible points and subsequently deducing the temperature variations at each of these points via a data processing algorithm. To actualize this methodology, a total of 80 temperature sensors were deployed to record the fluctuations in water temperature at 80 discrete locations within the tank body. Given that each individual intelligent inspection device was only capable of concurrently measuring temperatures at a maximum of 31 points, three such intelligent inspection devices were incorporated into the setup. The specific locations of these measuring points are elaborated in the following section.

The irregular structure of the pulse combustor causes the internal water areas in the steam boiler to be complex. We therefore divided the water from the combustion chamber inlet end to the tail pipe outlet into four parts, as shown in Fig 2a. In the z direction, the water area was divided into sections AB, BC, CD, and DE. Section AB was the combustion chamber, section BC was the part between the combustion chamber and the tailpipe, section CD was the spiral part of the tail pipe, and section DE was the part between the tail pipe and the decoupling chamber. Temperature sensors were placed on the intermediate planes perpendicular to the z-axis, i.e. the a–a plane, b–b plane, and e–e plane from section AB, section BC, and section

**Table 1. Temperature sensor technical parameters.**

| Parameters | Size |
| --- | --- |
| Model | WZPM-201 |
| Type | Pt100 |
| Measuring range/(°C) | −150 to 200 |
| Accuracy class | A |
| Length/(mm) | 30 |

**Table 2. Intelligent inspection device technical parameters.**

| Parameters | Size |
| --- | --- |
| Model | XMD-2000A31 |
| Channels | 31 |
| Sampling time/(ms) | 100 |
| Input power/(V) | 85–220 |
| Baud rate/(bps) | 2400–9600 |
| Precision/(%Fs) | 0.2 |
| Interface | USB 2.0 |
| Communication protocol | RS485 MODBUS-RTU |

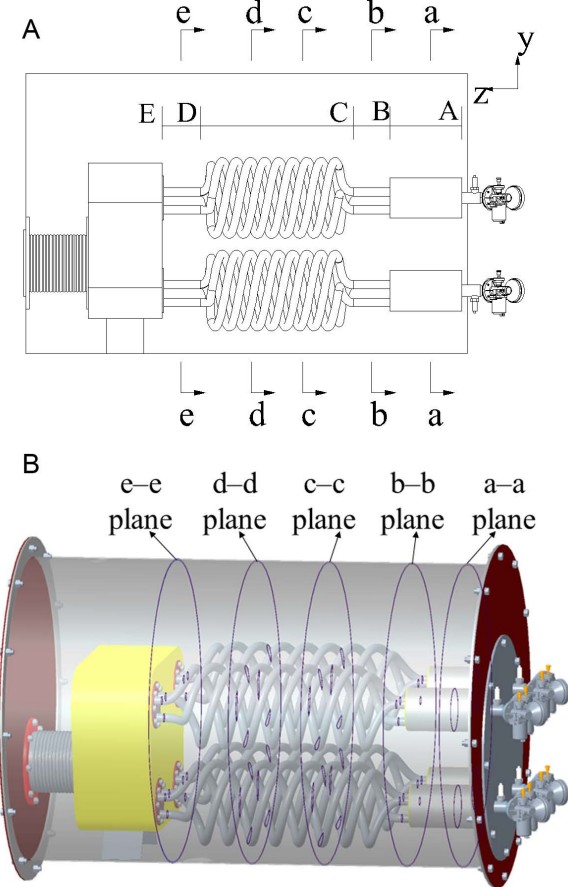

**Fig 2. Water area sections.** (a) Five cross-sections in a planar state. (b) Five sections in a three-dimensional state.

DE, respectively. Section CD was long, so it was divided into three equal parts to give a c–c plane and d–d plane, on which temperature sensors were placed, to improve the measuring point distribution. Five sections in a three-dimensional state are shown in Fig 2b.

Given the symmetrical characteristic of the water area inside the tank body, the temperature measuring points were disposed at one side of the water zone. Each temperature measuring point was meticulously labelled following the format of plane letter-position number. It is noteworthy that, for the temperature measuring points with the same position number in every plane, their x-axis and y-axis coordinates were identical. The locations of the temperature sensors within each plane, as well as the comprehensive temperature measuring system, are depicted in Fig 3.

Temperature measuring points 1, 9, 12, 14, and 16 were along the diameter line along the y-axis of each plane, making it easy to study temperature changes along the y-axis. Temperature measuring points 2, 3, 4, 13, and 15 were in a circle around pulse combustor no. 1 in the combustion chamber and temperature measuring points 5, 6, 7, 8, 10, and 11 were in a circle around pulse combustor no. 3 in the combustion chamber. This allowed temperature changes around the pulse combustor to be determined. The diameter of each circle was 20 mm more than the outer diameter of the spiral tail pipe.

Temperature measuring points 4, 14, and 15 were on a line connecting the axes of pulse combustors no. 1 and 2 to allow interactions between the pulse combustors to be investigated.

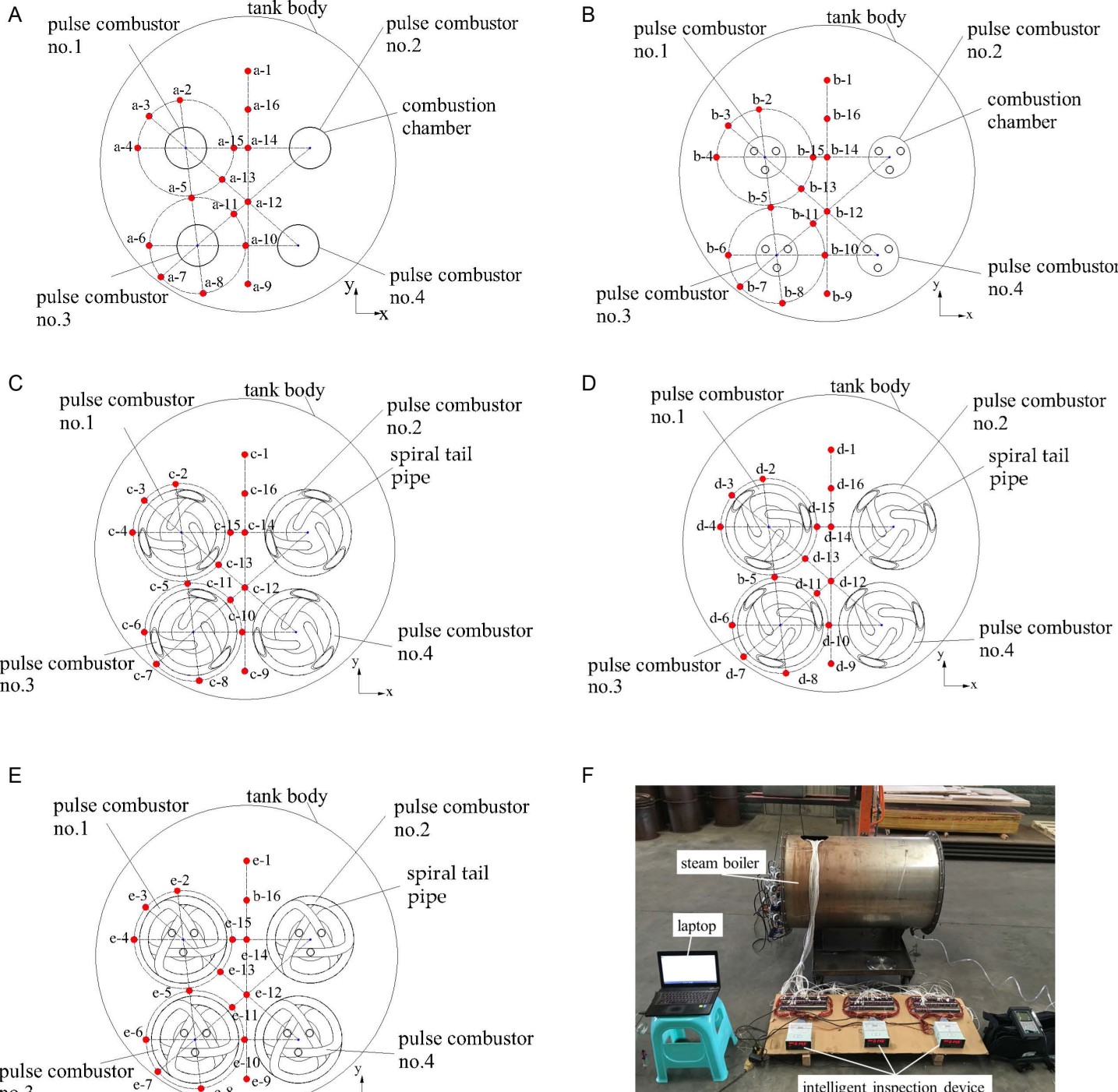

**Fig 3. Positions of the temperature measuring points and the temperature measuring system.** (a) a–a plane, (b) b–b plane, (c) c–c plane, (d) d–d plane, (e) e–e plane, and (f) temperature measuring system.

Temperature measuring points 2, 5, and 8 were on a line connecting the axes of pulse combustors no. 1 and 3. Temperature measuring points 3, 12, and 13 were on a line connecting the axes of pulse combustors no. 1 and 4. Temperature measuring points 7, 11, and 12 were on

a line connecting the axes of pulse combustors no. 2 and 3. Temperature measuring points 6 and 10 were on a line connecting the axes of pulse combustors no. 3 and 4.

## 2.3. Experimental design and statistical analysis

Room-temperature water was initially introduced into the tank body, following which the steam boiler was activated. The alterations in water temperature were precisely detected by the array of temperature sensors and concurrently recorded by a laptop computer. Once the temperature at each of the measuring points attained 100 °C, the heating process was sustained for an additional 5 minutes before the experiment was terminated.

**2.3.1. Voronoi diagram.** The Voronoi diagram was developed by Dirichlet and Voronoi [34,35]. In a Voronoi diagram, a region containing a finite set of discrete points is divided into convex polygonal subregions. Voronoi diagrams commonly use an unstructured mesh, also called Tyson polygons, and a Dirichlet mesh. The general mathematical description of a Voronoi diagram in any n-dimensional space is: let $\Omega$ be a convex space on any dimensional space $R_n$ and let $\{x_i\}$ be a set of arbitrarily distributed finite nodes in $\Omega$ ($x_i \in \Omega$, $i, j = 1, 2,…n$, where $i \neq j$ and $x_i \neq x_j$), then subspace $T_i$ can be defined for each node $x_i$, satisfying the requirements.

$$T_i = \left\{ x \,|\, d(x,x_i) < d(x,x_j), x \in \Omega, \forall j \neq i \right\} \tag{1}$$

where d is the distance (mm) and $T_i$ is the Voronoi cell of node $x_i$. $T_i$ represents the set of spatial positions of discrete points closest to node $x_i$. Obviously, $x_i \in T_i$. When the set of finite points $\{x_i\}$ is determined, the partition of $\{T_i\}$ is also determined and the partition is unique.

Taking a two-dimensional plane as an example (Fig 4), if there are discrete distributed and non-overlapping finite point sets $\{p_i\}$ on the plane, the plane has the unique Voronoi cell $\{V_i\}$. $\{V_i\}$ is an open convex polygon with a boundary determined by the vertical bisector of node $p_i$ and the adjacent node $p_j$. The resulting grid is a Voronoi diagram. The triangle obtained by connecting the central nodes to the common Voronoi element boundary is called the Delaunay triangle, and the triangle mesh obtained is called Delaunay triangulation.

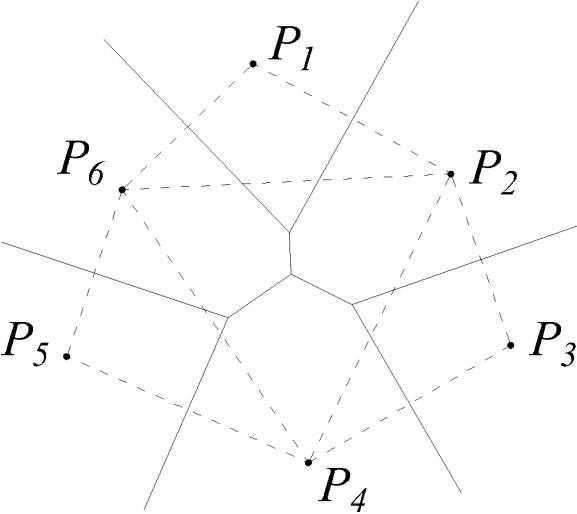

**Fig 4. Voronoi diagram and Delaunay triangulation.** p1, p2, .. , p6 are discrete points in the area. The solid lines represent the Voronoi diagram and the dotted lines represent Delaunay triangulation.

**2.3.2. Natural neighbor interpolation.** The natural neighbor interpolation method is an interpolation method based on the Voronoi diagram. The method calculates interpolation results for an interpolation point from the contributions of the natural neighbors to the function values of the different interpolation points. The interpolation format is:

$$f(x) = \sum_i^N \phi_i(x) f_i, x \in \Omega, i \in (1,2,3,\cdots\cdots N) \tag{2}$$

where *f(x)* is the interpolation result for point x, $i$ is the number of the natural neighbor of the node, fi is the physical quantity value for node $x_i$, and $\phi_i(x)$ is the interpolation function for node $x_i$.

Research into solving interpolation functions has been performed. Constructing an interpolation function involves inserting interpolation point x as a new node in the original Voronoi diagram and then generating a new updated Voronoi diagram. The area of the Voronoi cell containing interpolation point x can be labeled $S_x$, then $S_x$ can be divided into N parts using the original Voronoi diagram. The area of each part will be $S_1$, $S_2$, $S_3$, ... , $S_N$. The interpolation function can be defined as the proportions these parts contribute to the total area, as shown below.

$$\phi_i(x) = \frac{S_i}{S_x}, i \in (1,2,3,\cdots\cdots N) \tag{3}$$

In Fig 5, interpolation point $P_7$ is inserted into the original Voronoi diagram to give a new Voronoi diagram. The solid lines represent the Voronoi diagram and the dotted lines represent Delaunay triangulation. The area of the Voronoi unit containing $P_7$ is labeled $S_7$. $S_7$ is divided into six parts by the original Voronoi diagram with areas $S_1$, $S_2$, $S_3$, ... , $S_6$. The interpolation result for point $P_7$ is

$$f(P_7) = \sum_i^6 \phi_i(P_7) f_i, i = 1,2,3,\cdots\cdots 6 \tag{4}$$

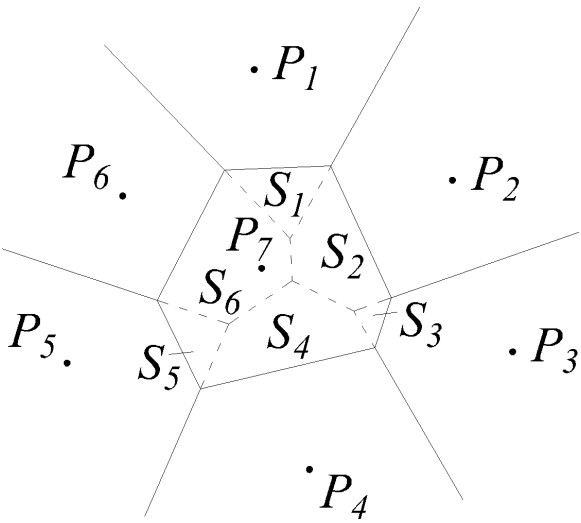

**Fig 5. Voronoi diagram and Delaunay triangulation after adding interpolating node P7.**

Where,

$$\phi_i(P_7) = \frac{S_i}{S_7}, i = 1, 2, 3, \cdots\cdots 6 \tag{5}$$

## 3. Results

The modelled temperature field from the start time to 35 min is shown in Fig 6.

The temperature of the whole system was relatively low at the start. At 5 min, the temperature of the water area near the pulse combustors, particularly near the combustion chamber and tail pipe, had increased rapidly. Some of the water areas were close to or at boiling point. However, the water far from the pulse combustors had not markedly changed temperature. There was a marked temperature gradient in the water area.

At 10 min, the area of high-temperature water was larger in the c–c and d–d planes than in other planes because of the large heat transfer area of the tail pipe. The temperatures of most of the water areas, including the areas far from the pulse combustors, continued to increase and only the temperature of a small part of the water at the bottom of the tank body remained unchanged.

At 15 min, the heat generated by the pulse combustors had increased the temperature of the whole water area. The temperatures of most of the water areas had increased to >45 °C and the area of high-temperature water near the tail pipe continued to expand.

At 20 min, the temperature of the whole water area continued to increase and the temperatures of most of the water areas had increased to >56 °C. The temperature of the water area at the bottom of the barrel still increased slowly and the strongest temperature gradients were mainly in this area.

At 25 min, the temperatures of most of the water areas in the c–c and d–d planes had reached 89 °C and the areas of high-temperature water near the pulse combustors in the a–a, b–b, and e–e planes had expanded rapidly.

At 30 min, the temperature of most of the water in the tank body had increased to >89 °C. The data indicated that saturated steam started to be generated and only the temperature of a small area of water at the bottom of the cylinder was increasing slowly.

At 35 min, the overall water temperature had increased to >89 °C.

The temperature distribution changes at a-a plane along the y-axis are shown in Fig 7.

It can be seen from Fig 7 that the temperature at each point increased over time. At the start, the temperature at each point was relatively low (~20 °C), but by 35 min the temperatures at all points had reached 100 °C. After examining the temperature distribution at each point at each time using the dotted line shown in Fig 7 as the boundary, the points were divided into three areas along the y-axis moving in the positive direction. These areas were labeled Area 1, Area 2, and Area 3.

Area 1 was at the bottom of the steam boiler. During heating, the temperature gradient was stronger in Area 1 than the other areas. The temperature gradient was strongest at 30 min. The maximum temperature in Area 1 was 95.6 °C, the minimum temperature was 34.3 °C, and the temperature difference was 61.3 °C. There was a temperature distribution peak near y = −220 cm. The temperature distribution along the y-axis in the positive direction first increased and then decreased. The changes were most clear at 5 min, when the amplitude was 12.3 °C. Over time, the changes gradually weakened and the amplitude was ~0 °C at 30 min. This would mainly have been because pulse combustors no. 3 and 4 were near y = −220 cm. Convective heat transfer to the water between these pulse combustors would have been relatively intense and the water temperature would have increased faster in this area than

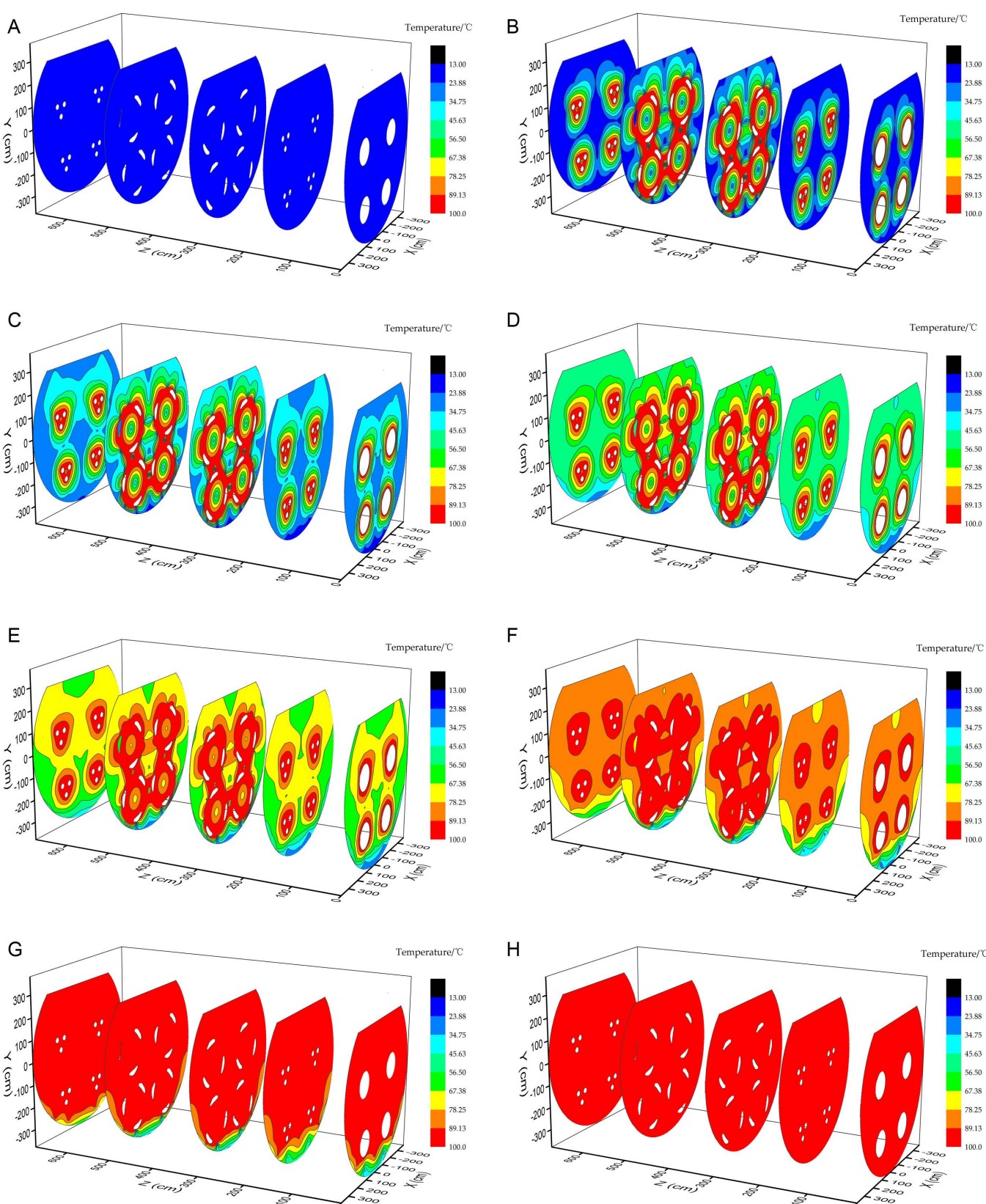

**Fig 6. Modeled temperature field.** (a) At the start, (b) at 5 min, (c) at 10 min, (d) at 15 min, (e) at 20 min, (f) at 25 min, (g) at 30 min, and (h) at 35 min.

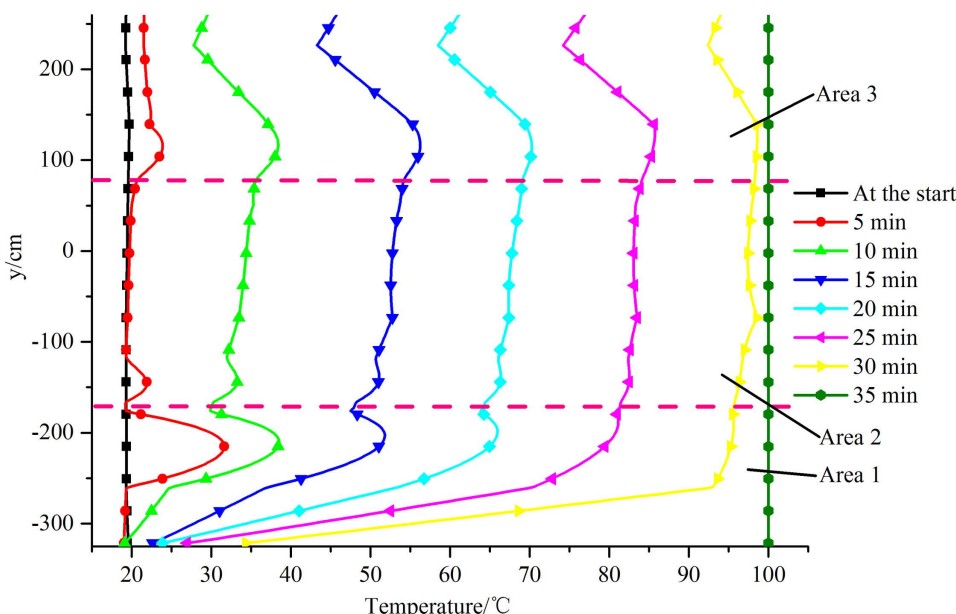

**Fig 7. Temperature distributions on the diameter line along the y-axis.**

above and below the area. This would have resulted in there being a peak in the temperature distribution near y = −220 cm.

Area 2 was in the middle of the steam boiler. The temperature distribution in Area 2 changed gently at all times. The temperature gradually but not strongly increased along the y-axis in the positive direction. The maximum increase was 6.1 °C at 15 min. The main reason for this would have been that the area was far from the four pulse combustors. During heating, heat would have been transferred to Area 2 mainly through natural convection. The heat transfer intensity at each point was not strong, and the differences between the heat transfer intensities at the different points were not marked. The temperature therefore changed relatively gently.

Area 3 was at the top of the steam boiler. The temperature distribution was more complex in Area 3 than the other areas. From the bottom of Area 3 along the y-axis in the positive direction, the temperature distribution first gradually increased and reached a maximum near y = 115 cm, then started to decrease and reached a minimum near y = 225 cm, and then started to increase again. The maximum temperature change occurred at 15 min, when the amplitude was 13.0 °C. The maximum value was the same as for Area 1. This would have been because pulse combustors no. 1 and 2 were near y = 115 cm. Convective heat transfer would have been stronger for the water in this area than above and below the area, so the temperature increased rapidly. The reason for the minimum value is shown in Fig 8 focusing on the water flow distribution in the area y > 0. During the heating of the steam generator, heat transfer to the internal water would mainly have been through natural convection. Water in different locations would have been at different temperatures and therefore would have had different densities, meaning some areas would have been buoyant relative to other areas. The hotter water would have flowed upward and the cooler water would have flowed downward. The water temperature above pulse combustors no. 1 and 2 would have increased rapidly, causing the water to flow upward and then disperse left and right after reaching the water surface. The two flow branches would have met, merged, and move downward in the middle of the water surface. The water temperature between pulse combustors no. 1 and 2 was relatively high,

resulting in vertical upward movement of water and convergence with downward water flow near the water surface, forming a mixing region. The intersection area would have been where the water arrived latest, so the temperature there increased slowly and the water temperature in that area was lower than in the upper and lower areas, resulting in the appearance of an upper temperature distribution minimum.

It can also be seen from Fig 7 that the area inside the steam boiler in which the water temperature increased slowest was at the bottom of the tank body. This will be useful information for structural optimization of subsequent units.

The temperature changes along the straight line through temperature measuring points a-4, a-14, and a-15 shown in Figs 4a and 9.

The centers of the combustion chambers in pulse combustors nos. 1 and 2 were at x = −135 cm and x = 135 cm, respectively, and there was no water near these locations. It can be seen from Fig 9 that the temperature distribution at each time had a symmetrical distribution

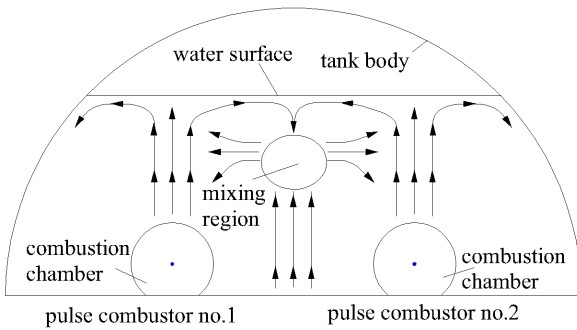

**Fig 8. Water flow distribution.**

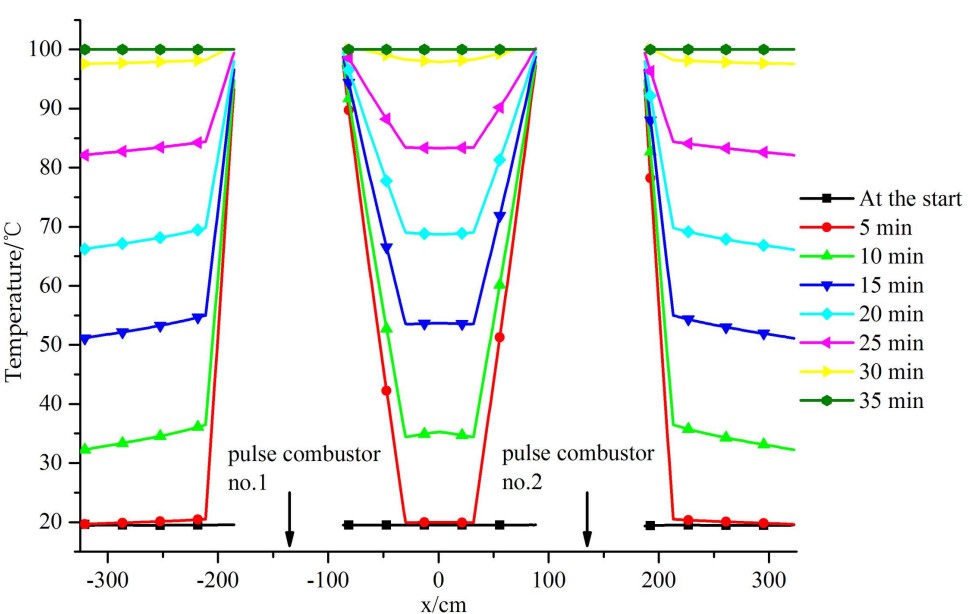

**Fig 9. Temperature distributions along the line through temperature measurement points a-4, a-14, and a-15.**

with an axis of symmetry at x = 0. This would mainly have been because the steam boiler was symmetrical. The temperature at each point on the straight line increased gradually over time. The outer wall of the combustion chamber of each pulse combustor was at a high temperature, so the temperature of the water area near the outer wall of the combustion chamber reached the corresponding saturation temperature at 5 min and the temperature gradually decreased in the direction away from the combustion chamber, giving an M-shaped distribution. At 5 min, the strongest temperature gradient was along the straight line. The water temperature near the outer wall of each combustion chamber was 100 °C but the water temperature near the inner wall of the steam generator was still the same as the initial temperature (~20 °C), meaning the temperature difference was 80 °C. Over time, the temperature gradient along the straight line became weaker, and by 35 min the temperature difference was 0 °C.

## 4. Discussion

Pulse combustion, a promising variant of combustion, is characterized by its non-stationary nature and pulsating behaviour, distinguishing it from conventional combustion methods. In recent years, it has garnered significant attention from researchers [36–39]. During the pulse combustion process, there are two fundamental excitation processes involved: under certain conditions, combustion can trigger acoustic pulsations within the burner; conversely, these acoustic pulsations can in turn alter the characteristics of combustion. Pulse combustion is formed by the positive mutual feedback of these two excitation processes under specific circumstances. It is precisely this special mutual feedback in pulse combustion that can significantly reduce the thickness of the heat transfer boundary layer. Meanwhile, it greatly enhances the processes of heat transfer, mass transfer, and momentum transfer between particles as well as between particles and gas flow within the combustion chamber and the tailpipe. Consequently, the pulsating burner possesses unparalleled advantages compared to conventional stable burners. The main distinguishing parameters are presented in Table 3 [40].

The comparison between pulse combustion and steady combustion reveals several notable differences in key performance parameters. Pulse combustion generally exhibits much higher combustion intensity (up to 50,000 kW·m$^{-3}$) and combustion efficiency (up to 99%) compared to steady combustion (100–1,000 kW·m$^{-3}$ and 80–96%, respectively). Additionally, due to the weakening effect of pulse combustion on the heat transfer boundary layer, it demonstrates a superior heat transfer coefficient (up to 500 W·m$^{-2}$·K$^{-1}$), enhancing its ability to transfer heat more efficiently. The excess air coefficient for pulse combustion is slightly lower, indicating a more optimal air-fuel ratio. Furthermore, pulse combustion results in lower CO emissions (0–1%) and significantly reduced NOx emissions (20–70 mg·m$^{-3}$) compared to steady combustion, making it more environmentally friendly. However, due to the mutual excitation between acoustics and combustion process required for pulse combustion, it generates higher noise levels (up to 130 dB), which may limit its application in noise-sensitive environments.

Table 3. Parameters comparison of pulse combustion and steady combustion.

| Combustion parameter | Steady combustion | Pulse combustion |
|---|---|---|
| Combustion intensity (kW·m$^{-3}$) | 100–1000 | 10000–50000 |
| Combustion efficiency (%) | 80–96 | 90–99 |
| Heat transfer coefficient (W·m$^{-2}$·K$^{-1}$) | 50–100 | 100–500 |
| Excess air coefficient | 1.01–1.2 | 1.00–1.01 |
| CO emission concentration (%) | 0–2 | 0–1 |
| NO$_x$ emission concentration (mg·m$^{-3}$) | 100–7000 | 20–70 |
| Noise (dB) | 85–100 | 110–130 |

Although pulse combustion has the advantages mentioned above compared with the conventional steady combustion, it also has obvious disadvantages. Besides the high noise level described in Table 3, it also has extremely high requirements for working conditions. For a pulse combustor to achieve pulse combustion, the acoustic conditions must be coupled with the heating conditions so that the gas flow inside the pulsating burner can oscillate. The acoustic conditions refer to the combustion chamber (with volume V) and the nozzle (with length L and inner diameter d) of certain dimensional parameters, while the heating conditions refer to the combustible mixture formed by a certain amount of fuel (Q1) and air (Q2). Since many working mechanisms of pulse combustion remain unclear to date, a certain acoustic structure has a specific acoustic resonant frequency, but it may not necessarily match the heating conditions or even form an effective coupling with any heating conditions, that is, the pulse combustion oscillation relationship cannot be formed. Therefore, only the acoustic conditions of certain specific structural dimensions can match the heating conditions within a certain range to produce a coupling effect, that is, the gas flow inside the pulsating burner works at a certain coupling oscillation frequency ($f$). When the coupling relationship exists, not only changing the dimensional parameters V, L, and d in the acoustic conditions will change the oscillation frequency ($f$), but also changing Q1 and/or Q2 in the heating conditions while keeping the acoustic conditions unchanged will change the coupling relationship, that is, change the oscillation frequency ($f$) and form a new pulsating oscillation system [41].

Therefore, for a pulse combustor, the current biggest challenge is how to design the structural dimensions with a stable working frequency based on its working environment (such as in air or in water, because this involves heat exchange and heating conditions). Unfortunately, there are currently no mature theoretical formulas available for calculation, and most of the structural design research is carried out using the trial-and-error method. That is, a series of structural dimensions are designed to see which one has a stable working frequency under specific working conditions.

Table 4 presents a comprehensive comparison of several steam boiler models, including those from the United States, Germany and this study. The parameters considered are crucial for evaluating the performance and characteristics of these devices. There is a positive correlation between parameters such as fuel consumption, steam production, empty weight, power and the dimension. That is, the larger the dimension, the greater the values of fuel consumption, steam production, empty weight, and power. For example, the S-2000 model has the largest product dimension, and correspondingly, it also has the largest fuel consumption, steam production, empty weight, and power. The steam boiler designed in this study has a volume dimension larger than only the SF-11 model and smaller than all other models. It also has the lightest empty weight, which implies that it can hold a greater amount of water.

**Table 4. Comparison of several steam boiler models.**

| Model | Dimension:length (m) × breadth (m) × height (m) | Fuel consumption (L·h⁻¹) | Steam production (kg·h⁻¹) | Empty weight (kg) | Power (kW) | Product origin | Affiliation company | $FCR$(L·kg⁻¹) |
|---|---|---|---|---|---|---|---|---|
| SF-11 | 1.5 × 1.0 × 1.3 | 11.6 | 167.8 | 407.9 | 125.1 | the United States | Sioux | 0.06913 |
| SF-20 | 1.8 × 1.3 × 1.5 | 21.4 | 308.2 | 543.8 | 231.8 | the United States | Sioux | 0.06944 |
| MS-100 | 2.2 × 1.0 × 1.65 | 8.5 | 100 | 600 | 74 | Germany | MSD AG | 0.08500 |
| MS-200 | 2.6 × 1.1 × 2.0 | 20 | 250 | 900 | 185 | Germany | MSD AG | 0.08000 |
| S-250 | 13.0 × 12 × 1.8 | 21 | 300 | 1200 | 220 | Germany | MSD AG | 0.07000 |
| S-750 | 4.1 × 1.5 × 2.5 | 64 | 800 | 2300 | 580 | Germany | MSD AG | 0.08000 |
| S-2000 | 5.7 × 2.1 × 3.0 | 174 | 2000 | 4200 | 1500 | Germany | MSD AG | 0.08700 |
| This study | 1.4 × 1.2 × 1.2 | 10.3 | 150 | 300 | 110.8 | —— | —— | 0.06867 |

Compared with the SF-11 model, it has a lower power and steam production, but at the same time, its fuel consumption is also lower than that of the SF-11 model.

To better analyse the relationship, the fuel consumption rate per unit steam production for each model is calculated, which is modelled as follows:

$$FCR = \frac{\text{fuel consumption}}{\text{steam production}} \tag{6}$$

FCR can, to a certain extent, represent the fuel utilization efficiency. A lower FCR indicates more efficient fuel utilization for steam production. There is no simple linear relationship across all models. The FCR values range from 0.06867 L·kg$^{-1}$ for the model in this study to 0.08700 L·kg$^{-1}$ for the S-2000 model. The boiler in this study exhibits relatively favourable FCR performance compared to several other models, suggesting that it has a relatively efficient conversion of fuel to steam. Besides, the volume of boiler in this study is 861L.

## 5. Conclusion

This research was dedicated to the design and in-depth analysis of a steam boiler that employs Helmholtz-type pulse combustors for soil steam disinfection (SSD) within horticultural greenhouses. The scope of this work encompassed a meticulous engineering design process of the boiler, integrating multiple pulse combustors and a carefully configured water flow and heat transfer system. To comprehensively investigate the temperature field evolution within the boiler, an elaborate experimental setup was established, which incorporated 80 precisely positioned temperature sensors. Subsequently, the Natural Neighbor Interpolation method was applied to model and analyze the collected data.

This research has yielded several significant achievements. Initially, we successfully elucidated the dynamic characteristics of the heating process. It was observed that in the early stages of operation, the temperature in the proximity of the pulse combustors exhibited a rapid increase, while regions situated at a greater distance from the heat source demonstrated a relatively slower heating rate. After 35 minutes of operation, the majority of the water within the boiler attained temperatures exceeding 89 °C, indicating the efficient generation of saturated steam. Moreover, the identification of temperature gradients along various axes and planes furnished critical insights for potential structural optimization of the boiler. Secondly, in comparison with existing steam boiler models, the boiler designed in this study exhibited a relatively compact volume and a lightweight empty weight. Notably, it demonstrated a highly competitive fuel consumption rate per unit steam production, registering at 0.06867 L·kg$^{-1}$, thereby highlighting its enhanced energy efficiency.

This work makes a substantial contribution to the field of SSD by introducing a novel boiler design with improved performance attributes. Future research endeavors will be predicated on these findings, aiming to further optimize the design of the pulse combustor and to explore the long-term performance and practical application potential of the boiler in a diverse range of horticultural scenarios.

## Supporting information

**S1 File. Raw data.**
(RAR)

## Author contributions

**Conceptualization:** Sipu Pan.
**Data curation:** Sipu Pan.

**Formal analysis:** Sipu Pan.

**Funding acquisition:** Sipu Pan.

**Investigation:** Sipu Pan.

**Methodology:** Sipu Pan.

**Project administration:** Sipu Pan.

**Resources:** Sipu Pan.

**Software:** Sipu Pan.

**Supervision:** Sipu Pan.

**Validation:** Sipu Pan.

**Visualization:** Sipu Pan.

**Writing – original draft:** Sipu Pan.

**Writing – review & editing:** Sipu Pan.

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
