## [Decision Letter · Decision Letter 0]

5 Nov 2024

PONE-D-24-38861Data Collection and Analysis of Temperature Field Simulation of Steam Boiler for Soil Steam DisinfectionPLOS ONE

Dear Dr. Pan,

Thank you for submitting your manuscript to PLOS ONE. After careful consideration, we feel that it has merit but does not fully meet PLOS ONE’s publication criteria as it currently stands. Therefore, we invite you to submit a revised version of the manuscript that addresses the points raised during the review process.

We look forward to receiving your revised manuscript.

Kind regards,

Soheil Mohtaram

Academic Editor

PLOS ONE

Journal Requirements:

2. Thank you for stating the following financial disclosure: "the Natural Science Foundation of Jiangsu Province (Grant no. BK20200142), the Natural Science Foundation of the Jiangsu Higher Education Institutions of China (Grant no. 22KJB210014), the Start-up Fund for New Talented Researchers of Nanjing Vocational University of Industry Technology (Grant no. YK-19-04-03)."

3. Thank you for stating the following in the Acknowledgments Section of your manuscript: "The authors acknowledge the Natural Science Foundation of Jiangsu Province (Grant no. BK20200142), the Natural Science Foundation of the Jiangsu Higher Education Institutions of China (Grant no. 22KJB210014), the Start-up Fund for New Talented Researchers of Nanjing Vocational University of Industry Technology (Grant no. YK-19-04-03)."

Please remove any funding-related text from the manuscript and let us know how you would like to update your Funding Statement. Currently, your Funding Statement reads as follows: "the Natural Science Foundation of Jiangsu Province (Grant no. BK20200142), the Natural Science Foundation of the Jiangsu Higher Education Institutions of China (Grant no. 22KJB210014), the Start-up Fund for New Talented Researchers of Nanjing Vocational University of Industry Technology (Grant no. YK-19-04-03)."

6. PLOS requires an ORCID iD for the corresponding author in Editorial Manager on papers submitted after December 6th, 2016. Please ensure that you have an ORCID iD and that it is validated in Editorial Manager. To do this, go to ‘Update my Information’ (in the upper left-hand corner of the main menu), and click on the Fetch/Validate link next to the ORCID field. This will take you to the ORCID site and allow you to create a new iD or authenticate a pre-existing iD in Editorial Manager.

Reviewers' comments:

Reviewer's Responses to Questions

**Comments to the Author**

1. Is the manuscript technically sound, and do the data support the conclusions?

Reviewer #1: Yes

Reviewer #2: Partly

2. Has the statistical analysis been performed appropriately and rigorously? 

Reviewer #1: Yes

Reviewer #2: Yes

3. Have the authors made all data underlying the findings in their manuscript fully available?

Reviewer #1: Yes

Reviewer #2: Yes

4. Is the manuscript presented in an intelligible fashion and written in standard English?

Reviewer #1: Yes

Reviewer #2: Yes

5. Review Comments to the Author

Reviewer #1: - The abstract should be improved (grammer and content). The goal of this article and content is not obvious.

- Please improve the writing and grammar of your manuscript.

- The conclusion should be improved. The author should explain the scope of work and achievement in a more clear way in the conclusion not in bullet wise.

- In the reference section, the author should use the newest articles.

- The captions font of figures and tables should be less than the another content of article.

- The author should revise the style and format of the references.

Reviewer #2: This article is devoted to investigation of the efficient way to generate water vapour for soil steam disinfection by using pulse combustors. The author claims that using the pulse combustors, that use spiral combustion chamber emerged into water, can help achieving better fuel efficiency than other types of water heaters to generate vapor. In the present work, this problem was investigated theoretically by numerical simulation with natural neighbour interpolation method and experimentally by placing temperature sensors inside the boiler and measuring temperature distribution.

While the work is interesting and topical, its novelty and significance are not directly clear from the manuscript. We would ask the author to try to further improve the presentation. Our particular comments for this follow.

1. One of the key claims of the present work relates to the advantage of pulse burner boilers. Is there any comparison of pulse burner boilers with other types of boilers (like the amount of fuel used to reach certain temperature difference for certain amount of water)? Maybe it is worthwhile to add this in the manuscript in more detail, with numbers, to make this more informative for readers?

2. Can other water vapor applications other than Soil Steam Disinfection take advantage from pulse combustors boilers? It would be interesting to know in view of current detailed temperature field simulation and measurements.

3. What are the disadvantages of pulse combustors in comparison with other types of combustors? It is desirable to add this discussion in relation to applications such as the disinfection.

4. What is the volume of boiler, weight of boiler, heating power, fuel consumption and how it compares with other boilers? It is desirable to know these details from the manuscript. Maybe, in the form of a table?

5. Most importantly, the work is devoted to the temperature field simulation, however it is not clear from the manuscript how the simulation was done. Which software was used for the numerical simulation? How this was used, with which boundary conditions, which equations and with which parameters solved?

6. Is there any dependence of efficiency of pulse combustion on pulse frequency? This seems to be important characteristics.

7. Please explicitly tell which graphs correspond to calculations and which ones to measurements. Were these data (obtained by calculations and measurements) in agreement? Which graph or data demonstrate this?

6. PLOS authors have the option to publish the peer review history of their article (what does this mean? ). If published, this will include your full peer review and any attached files.

**Do you want your identity to be public for this peer review?** For information about this choice, including consent withdrawal, please see our Privacy Policy .

Reviewer #1: No

Reviewer #2: No

---

## [Author Response · Author response to Decision Letter 1]

13 Jan 2025

Response to Reviewers

Journal Requirements:

1. I have made formatting changes to the manuscript according to the PLOS ONE's style requirements.

2. About “Financial Disclosure”, I would like to change it to the following content:

This paper is funded by the Natural Science Foundation of the Jiangsu Higher Education Institutions of China (No. 22KJB210014), the Start-up Fund for New Talented Researchers of Nanjing Vocational University of Industry Technology (No. YK-19-04-03). The funders had no role in study design, data collection and analysis, decision to publish, or preparation of the manuscript.

3.I have removed all the funding information in the Acknowledgments Section of my manuscript. I would like to update my Funding Statement as follows: “the Natural Science Foundation of the Jiangsu Higher Education Institutions of China (No. 22KJB210014), the Start-up Fund for New Talented Researchers of Nanjing Vocational University of Industry Technology (No. YK-19-04-03)”.

4.With sincere dedication and anticipation for this resubmission, I would like to further explain an issue regarding the Data Availability Statement. Initially, I made a minor mistake as I did not fully comprehend its specific requirements. In the current submission, not all raw data were included. However, I have now packaged all the necessary raw data into a compressed file and uploaded it as Supporting Information files. This way, it should provide full access to the underlying data for those who may need it to reproduce or further explore my research findings.

5.I have updated my ORCID iD in the "Update my Information" as required.

6.This requirements is the same as the 5th.

I have updated my ORCID iD in the "Update my Information" as required.

Review Comments to the Author:

Reviewer #1

1. The grammar and content of the abstract have been improved, with a focus on the goal and the research content of this article. The revised abstract is as follows:

Soil steam disinfection (SSD) has emerged as a highly promising substitute for methyl bromide fumigation in the management of soil-borne pathogens, nematodes, and weed seeds. In the present study, an innovative steam boiler driven by Helmholtz-type pulse combustors was meticulously engineered to meet the requirements of SSD in horticultural greenhouses. The water within the boiler was partitioned into discrete zones, and a total of 80 temperature sensors were strategically positioned to precisely monitor the temperature fluctuations at specific locations. Leveraging the Natural Neighbor Interpolation method, a comprehensive model of the temperature field within the boiler was developed. The experimental findings demonstrated that at the initial stage, the temperature in the vicinity of the pulse combustors escalated rapidly, while the regions located farther away exhibited a relatively sluggish heating rate. As the heating process progressed, the area of high temperature expanded progressively. After 35 minutes of operation, the majority of the water within the boiler reached temperatures exceeding 89 °C, signifying the generation of saturated steam. Notably, distinct temperature gradients were discerned along different axes and planes, offering valuable insights for the structural optimization of the boiler. In comparison with other steam boiler models, the designed boiler boasted a relatively compact volume, a lightweight empty weight, and an impressively favorable fuel consumption rate per unit steam production, registering at 0.06867 L·kg⁻¹. These results unequivocally highlight the potential of this boiler for efficient SSD applications, thus laying a solid foundation for further research and development in this field.

2.About the writing and grammar, I have taken several meticulous steps. Firstly, I consulted with colleagues proficient in English writing, incorporating their suggestions to further polish the language. They carefully combed through the entire text, rectifying grammar errors, enhancing sentence structures, and ensuring coherence and flow throughout the paper. Secondly, I myself reread and revised the manuscript multiple times, paying close attention to word choice, punctuation, and overall readability. Through these combined efforts, I am confident that the current version of the manuscript exhibits a significantly improved standard of writing and grammar, making it more accessible and engaging for readers. I truly hope that these enhancements meet your expectations and contribute to the overall quality of the paper. Specific modifications can be found in the resubmitted version.

3.About the conclusion, I am truly grateful for the insightful comment regarding the improvement of the conclusion. I fully acknowledge the importance of presenting the scope of work and achievements in a more lucid and comprehensive manner, and I have made substantial revisions accordingly. I have rewritten the conclusion section to replace the previous bullet-point style. Instead, I now provide a cohesive and continuous narrative that elaborates on the overall scope of my research. I believe that this new version of the conclusion effectively conveys the essence of my research and hope it aligns with your expectations.

4.About references, I scoured through the latest publications in our field. I carefully selected the most pertinent and high-impact new articles, such as reference [4][36][37][40]. These newly added references have been integrated into the “Reference” section, supplementing and enriching the existing ones.

For instance, reference [40] published in 2024 expounds on the working principle of the Helmholtz-type pulsating combustor. It comprehensively reviews the current research and development trends of pulsating combustors and further summarizes its application in the field of pest control.

By doing so, I believe the reference section now reflects the most current state of knowledge in the area, providing readers with a more comprehensive and contemporary view of the subject matter.

5. I have adjusted the font settings for all figure and table captions to ensure they are visibly smaller than the main text of the article, as per the recommended standard. Specific modifications can be found in the resubmitted version.

6.About the style and format of the references, a reference management tool, EndNote, was used. Also, the “Plos” style file was used to assist me with the formatting of your references. I meticulously examined each reference within the manuscript. I made adjustments to the font, size, indentation, and alignment to precisely match the prescribed style. I also paid special attention to the citation style, correcting any discrepancies in the use of superscripts, parentheses, or numbered citations. Additionally, I double-checked for any potential errors or omissions, such as incorrect page numbers. Through these works, I am confident that the references now adhere strictly to the journal's style and format, enhancing the overall professionalism and readability of the manuscript.

Reviewer #2

1.I am very grateful for your astute observation and valuable suggestion regarding the comparison of pulse burner boilers with other types of boilers. You are right that adding such a detailed comparison would significantly enhance the informativeness and practical value of the manuscript. In response to this comment, I have conducted further research and analysis.

Firstly, the technical parameters between pulse combustion and conventional steady combustion were compared. The detailed parameter analysis can be found in the discussion section of the draft.

Secondly, in the discussion section, the technical parameters of the pulse burner boilers designed in this paper and those of other steady-state combustion steam boilers were also added. The compared products are from countries such as Germany and the United States. The specific parameters for comparison include product dimensions, fuel consumption, steam production, empty weight, power, FCR (the fuel consumption rate per unit steam production) and so on. From the Table 4 in the manuscript, we can say that the boiler in this study exhibits relatively favorable FCR performance compared to several other models, suggesting that it has a relatively efficient conversion of fuel to steam.

I believe that this addition not only strengthens my key claim about the advantages of pulse burner boilers but also provides readers with practical and actionable information that can be useful in real-world applications.

2. Pule combustion, as a unique unsteady combustion technology, exhibits special advantages in applications requiring intense heat and mass transfer due to the rapid reciprocating characteristics of its combustion exhaust gases. Specific application scenarios include the following:

(1) Material drying

Niu Haixia[1] took a study about mass transfer between materials and unsteady airflow from a Helmholtz type combustor. During the drying process, the dried material is placed inside the Tail pipe II.

(2) Pest control

During the working process of the pulse combustor, when the pressure inside the combustion chamber is greater than the atmospheric pressure, the pressure guiding pipe will introduce the high pressure into the medicine tank and open the medicine switch. Then, under pressure, the liquid medicine will flow from the medicine nozzle to the tailpipe. The high-temperature and high-speed exhaust gas discharged by the pulse combustor during its operation will cause the liquid medicine flowing into the tailpipe to crack, break up and evaporate into droplets.

(3)Steam boiler

Pulse combustors boilers have the potential to offer advantages in various water vapor applications beyond soil steam disinfection. In the food processing industry, for example, the unique temperature characteristics and high heat transfer efficiency of pulse combustors boilers could be utilized in processes such as pasteurization and sterilization. The rapid and precise temperature control achievable with these boilers may enhance the quality and safety of food products. Additionally, in the textile manufacturing sector, they could be employed for fabric dyeing and finishing operations, where controlled steam application is crucial for achieving consistent and high-quality results.

3.I would like to express my sincere gratitude for your valuable suggestion regarding the disadvantages of pulse combustors in comparison with other types. In response to this comment, I have made the following additions and improvements in the manuscript.

I have now included a detailed discussion on the drawbacks of pulse combustors. Firstly, it is noted that due to the unique mutual excitation mechanism between acoustics and the combustion process essential for pulse combustion, it generates relatively high noise levels, which can reach up to 130 dB. This significant noise emission may indeed pose limitations in noise-sensitive applications such as in certain indoor or residential settings where quiet operation is crucial.

Secondly, I have elaborated on the highly demanding working conditions required by pulse combustors. The need for precise coupling between acoustic conditions (determined by the specific dimensions of the combustion chamber and nozzle) and heating conditions (dictated by the fuel and air mixture) presents a major challenge. Given the current state of knowledge where many working mechanisms of pulse combustion remain elusive, achieving the right match between these conditions is often difficult. Even with a specific acoustic structure having its own resonant frequency, there is no guarantee of compatibility with the heating conditions, and in many cases, an effective coupling cannot be established, thereby preventing the onset of stable pulse combustion.

Furthermore, I have also addressed the current difficulties in designing pulse combustors. The absence of established theoretical formulas for calculating the optimal structural dimensions based on different working environments (such as in air or water) forces researchers to rely on the trial-and-error approach. This involves designing multiple sets of structural dimensions and testing them to identify the ones that can achieve a stable working frequency under specific conditions.

Specific modifications can be found in the “Discussion” section.

4. In response to your suggestions, Table.4 was added in “Discussion” section. The product dimensions, fuel consumption, steam production, empty weight, power, and FCR (the fuel consumption rate per unit steam production) of the boiler in this study, were compared with those of other boilers with the form of a table.

5. I sincerely appreciate your astute observation and the opportunity to clarify the methodology employed in the temperature field simulation. The temperature field simulation is carried out according to the following procedure:

(1) Temperature Data Collection

Temperature data were collected through 80 temperature measurement points during the process from the startup of the boiler until all the temperature measurement points reached 100 °C.

(2) Temperature Data Selection

Temperature data at eight specific moments (At the start, at 5th min, at 10th min, at 15th min, at 20th min, at 25th min, at 30th min, and at 35th min) from five cross-sections (a–a plane, b–b plane, c–c plane, d–d plane and e–e plane) were selected as the known conditions for the temperature field simulation, just like pi (i = 1,2,……6) shown in Figure 5 in the manuscript. These data can be found in the file named “temperature raw data.xlsx”.

(3) Temperature Field Simulation

Equation (4) was converted into a program that can be run in MATLAB software. The program files can be found in the file named “case1.m”.

(4) Simulation result output

Through program calculations, temperature data at all positions on each cross-section can be obtained, similar to the data of p7 shown in Figure 5 in the manuscript. Subsequently, Figure 6 in the manuscript can be obtained. The calculation results can be found in the file named “a-a plane.xlsx”

6.There is indeed a relationship between pulse combustion efficiency and pulse frequency. Let's assume that when the pulse frequency increases, the number of combustion events completed within a unit time also increases, and the number of times the combustion exhaust gas travels back and forth in the tailpipe also increases. This facilitates the timely expulsion of the exhaust gas from the previous combustion, providing sufficient combustion space for the new fuel-air mixture and ensuring that the new round of combustion is completer and more thorough, thereby improving the overall combustion efficiency.

In addition to the combustion efficiency increasing with the increase of the pulse frequency, the heat transfer efficiency also rises. This can be analyzed from the concept of the temperature boundary layer.

When a fluid with a temperature of t∞ flows over a wall surface with a wall temperature of tw (t∞＞tw), due to the influence of heat exchange, there will be an obvious change in temperature within the thin fluid layer close to the wall surface. Its characteristics are that on the wall surface, where y=0, t= tw, and at the position of the thin layer, that is, on the temperature boundary layer, where, y=δt，t= t∞. We call this thin fluid layer where the temperature changes significantly the temperature boundary layer.

The temperature boundary layer has the following characteristics: (1) The thickness δt of the temperature boundary layer is much smaller than the characteristic dimension of the object; (2) The temperature changes drastically along the thickness (y) direction within the boundary layer.

Due to the pulsation effect, the combustion exhaust gas repeatedly moves forward and backward in the tailpipe, which greatly reduces the originally stable boundary layer thickness δt, allowing more heat to approach the wall surface (y=0) and enhancing the heat transfer process. The higher the pulse frequency is, the stronger the weakening effect on the temperature boundary layer is, and therefore, the higher the heat transfer efficiency is.

7. I am very grateful for your detailed and perceptive comments, which have helped me to fur

---

## [Decision Letter · Decision Letter 1]

29 Jan 2025

Data Collection and Analysis of Temperature Field Simulation of Steam Boiler for Soil Steam Disinfection

PONE-D-24-38861R1

Dear Dr. Sipu Pan,

We’re pleased to inform you that your manuscript has been judged scientifically suitable for publication and will be formally accepted for publication once it meets all outstanding technical requirements.

Kind regards,

Soheil Mohtaram

Academic Editor

PLOS ONE

Review Comments to the Author

Reviewer #1: The comments on the article have been addressed, and the article can be published after the following comment:

In the conclusion section, instead of using the phrase "our research," please write in the third person as "in this research…."

Thank you.

Reviewer #2: The author has very carefully taken into account all the comments of the two Referees. We recommend its publication, after revisiting the figures. Text in Figs. 1(a), 2(b) and all others is so small that it is practically indistinguishable.

---

## [Editor Report · Acceptance letter]

PONE-D-24-38861R1

PLOS ONE

Dear Dr. Pan,

I'm pleased to inform you that your manuscript has been deemed suitable for publication in PLOS ONE. Congratulations! Your manuscript is now being handed over to our production team.

Kind regards,

on behalf of

Dr. Soheil Mohtaram

Academic Editor

PLOS ONE